# Fluorocarbonylation via palladium/phosphine synergistic catalysis

Mingxin Zhao[1,3], Miao Chen[1,3], Tian Wang[1], Shuhan Yang[1], Qian Peng [1,2] ✉ & Pingping Tang [1,2] ✉

Despite the growing importance of fluorinated organic compounds in pharmaceuticals, agrochemicals, and materials science, the introduction of fluorine into organic molecules is still a challenge, and no catalytic fluorocarbonylation of aryl/alkyl boron compounds has been reported to date. Herein, we present the development of palladium and phosphine synergistic redox catalysis of fluorocarbonylation of potassium aryl/alkyl trifluoroborate. Trifluoromethyl arylsulfonate (TFMS), which was used as a trifluoromethoxylation reagent, an easily handled and bench-scale reagent, has been employed as an efficient source of $COF_2$. The reaction operates under mild conditions with good to excellent yields and tolerates diverse complex scaffolds, which allows efficient late-stage fluorocarbonylation of marked small-molecule drugs. Mechanistically, the key intermediates of labile Brettphos-Pd(II)-OCF$_3$ complex and difluoro-Brettphos were synthesized and spectroscopically characterized, including X-ray crystallography. A detailed reaction mechanism involving the synergistic redox catalytic cycles Pd(II)/(0) and P(III)/(V) was proposed, and multifunction of phosphine ligand was identified based on $^{19}F$ NMR, isotope tracing, synthetic, and computational studies.

The introduction of fluorine atoms into organic molecules has attracted chemists' great attention over the years due to the enhanced physicochemical properties of the fluorinated compounds[1–7]. Acyl fluorides are of great importance in terms of synthetic utility, which are wildly applied in the solution and solid-phase peptide synthesis as the use of α-amino acid fluorides generates the corresponding amides in good yields without racemization[8–11]. Acyl fluorides are also commonly used in the transition-metal catalyzed cross-coupling reaction as the acylating reagents, fluorination reagents, and so on[12–15] (Fig. 1A). Synthesis toward acyl fluorides has been developed in the past few decades which were majorly categorized into two parts, nucleophilic deoxyfluorination of carboxylic acid[16–24] and Pd-catalyzed fluorocarbonylation of aryl halides with CO gas or equivalents[25–28]. However, the direct fluorocarbonylation of aryl/alkyl boron compounds has not been reported before. As the complementary method, herein, we present the palladium/phosphine synergistic redox catalysis to achieve fluorocarbonylation of potassium aryl/alkyl trifluoroborate

using trifluoromethyl arylsulfonate (TFMS) as $COF_2$ source through the synergistic redox of Pd(II)/Pd(0) and P(III)/P(V).

Due to the high corrosivity and toxicity, $COF_2$ is rarely studied[29–32] until now. Only a few cases showed the potential for the laboratory-scale synthesis of $COF_2$ through the fluoride substitution of (tri)phosgene[29,30,32] or decomposition of trifluoromethoxy anion and radicals[33–36]. The previous studies in our group suggested that trifluoromethyl arylsulfonate (TFMS) could generate trifluoromethoxy anions in situ, which could decompose into $COF_2$ and fluoride anions[37–39] (Fig. 1B). Inspired by the pioneering work of the Shreeve group, $COF_2$ could be easily reduced into CO and afforded the difluorophosphine as a byproduct[40] (Fig. 1B). Meanwhile, palladium-catalyzed fluorocarbonylation of aryl halides from CO source[25–28] and Lewis acidic phosphonium accelerated migratory insertion of CO[41,42] was well-documented. Taking these discoveries and the broad synthetic utilities of acyl fluorides into account, it is meaningful and promising to explore the palladium/phosphine

[1]State Key Laboratory and Institute of Elemento-Organic Chemistry, Nankai University, 300071 Tianjin, China. [2]Haihe Laboratory of Sustainable Chemical Transformations, 300192 Tianjin, China. [3]These authors contributed equally: Mingxin Zhao, Miao Chen. ✉ e-mail: qpeng@nankai.edu.cn; ptang@nankai.edu.cn

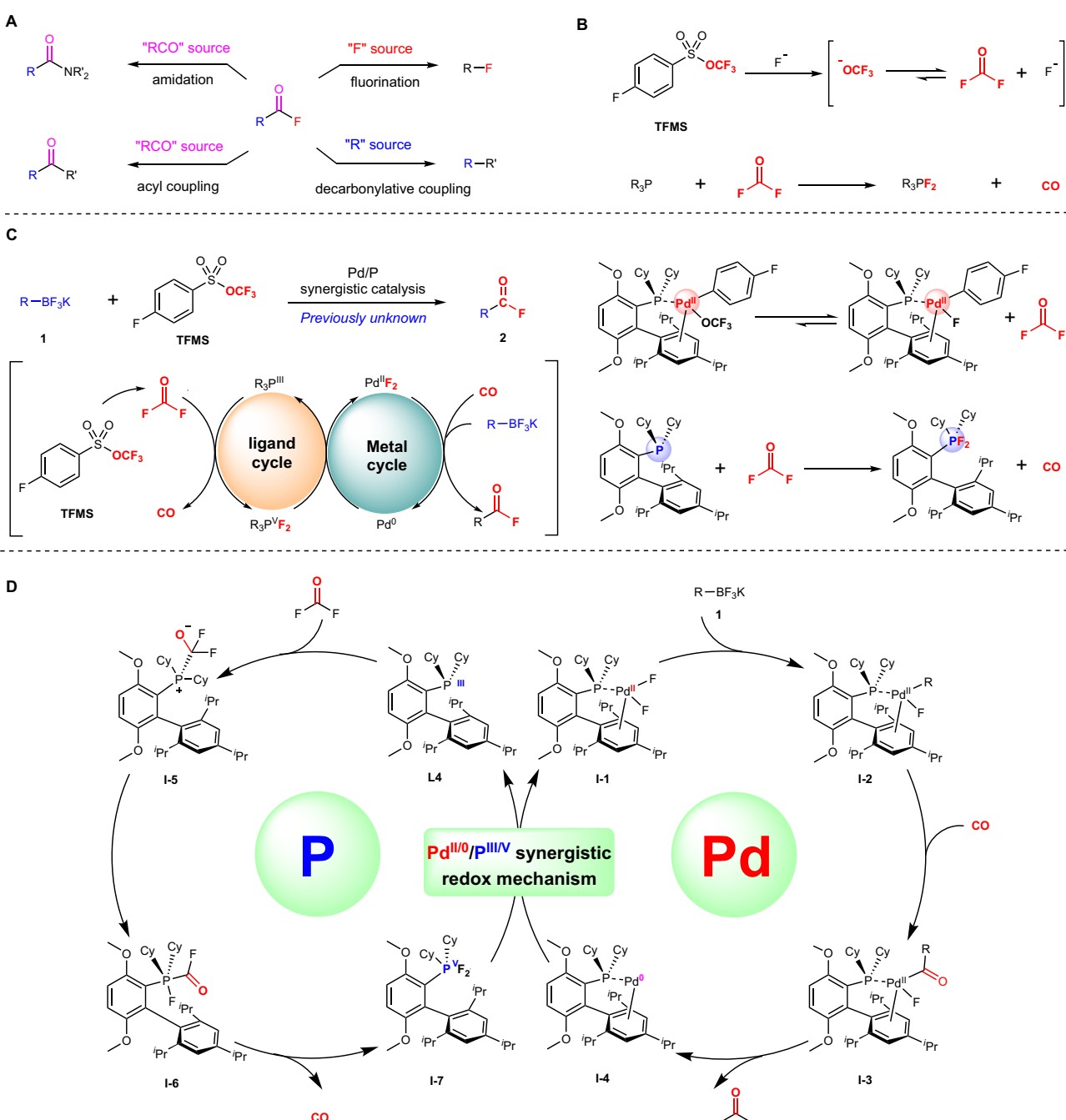

**Fig. 1 | Reaction design. A** Applications of acyl fluorides. **B** Decomposition of TFMS reagent and difluoro P(V) species and CO via the phosphine nucleophilic attack COF$_2$. **C** Our blueprint for palladium/phosphine synergistic redox catalysis of fluorocarbonylation of potassium aryltrifluoroborate. **D** Proposed mechanism.

synergistic redox catalysis fluorocarbonylation of potassium aryl/alkyl trifluoroborate with TFMS; we assumed that two interwoven catalytic cycles Pd(II)/(0) and P(III)/(V) might be engineered to generate simultaneously (1) LnArPd(II)F species and COF$_2$ via the β-F elimination of LnArPd(II)OCF$_3$ intermediate, and (2) difluoro P(V) species and CO via the phosphine nucleophilic attack COF$_2$. Through the different spatial coordination of fluoride ions, the reducibility of difluoro P(V) species may be tuned for matching the oxidative addition of LnPd(0) species to regenerate catalytic LnPd(II) species (Fig. 1C).

Our proposed mechanistic cycle for palladium/phosphine synergistic redox catalysis of fluorocarbonylation of potassium

aryl/alkyl trifluoroborate **1** is outlined in Fig. 1D. We presumed that the transmetallation of **1** would happen to form complex **I-2**, which underwent carbonyl insertion to generate intermediate **I-3**, followed by reductive elimination to release the product **2** and generate the Pd(0) species **I-4**. Concurrently with this palladium cycle, we envisioned that phosphine ligand **L4** reacts with COF$_2$ via nucleophilic attack to form **I-5**, followed by fluoride ion migration to afford five-coordinated P(V) intermediate **I-6**, which might proceed through CO release to afford activated difluoro P(V) species **I-7**. And then, a synergistic redox mechanism with Pd(0) species **I-4** may occur via ligand exchange and oxidative addition, regenerating catalyst Pd(II) species **I-1** and phosphine ligand **L4**.

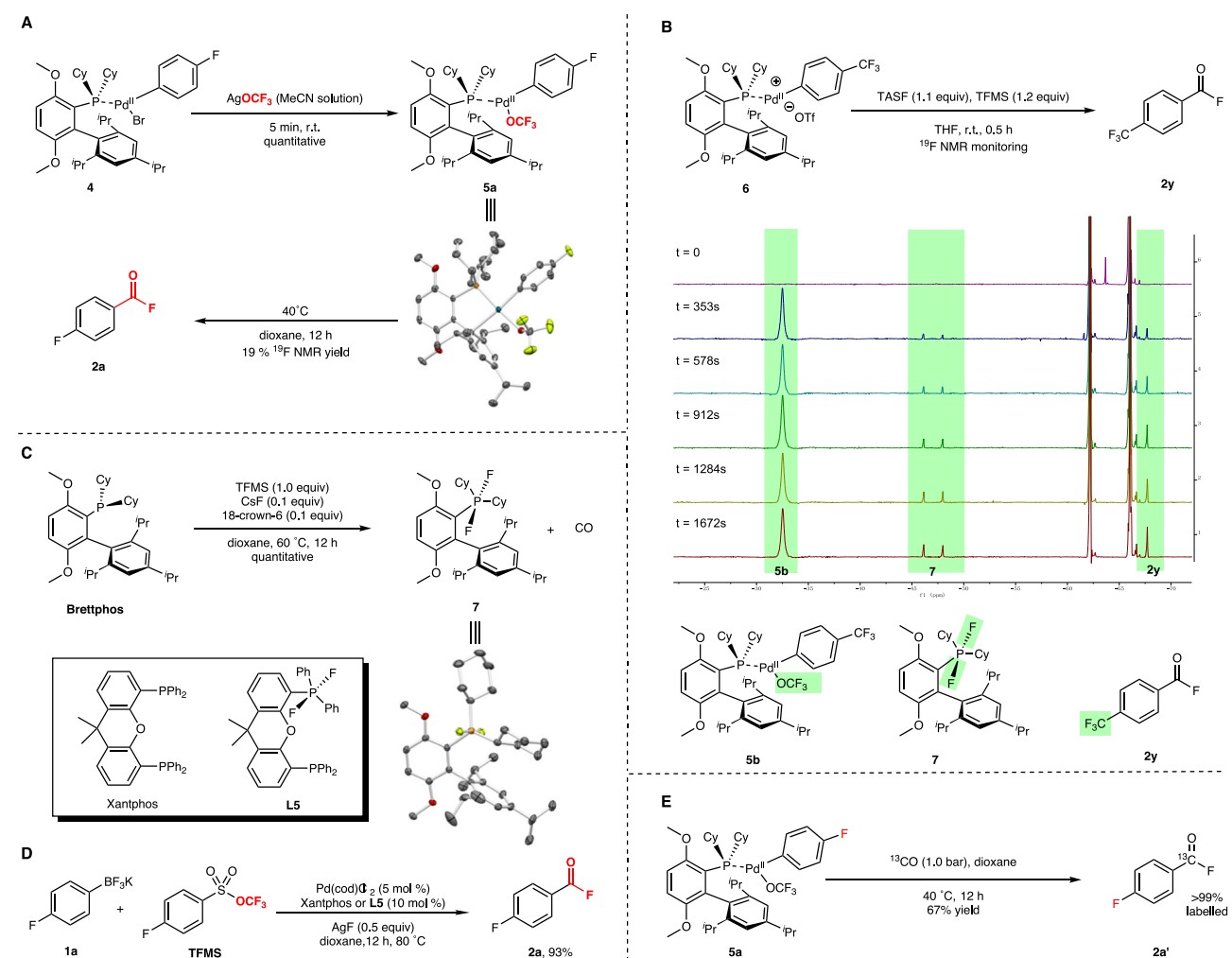

**Fig. 2 | Mechanism investigation. A** Synthesis of Pd(II)−OCF₃ complex. **B** Reaction course study. **C** Synthesis of difluoro-Brettphos. **D** Catalytic method for palladium/phosphine synergistic redox catalysis of fluorocarbonylation of potassium aryltrifluoroborate. **E** Isotope labeling of CO.

## Results

### Mechanistic investigation

The initial mechanism investigation was focused on stoichiometric experiments. Following Buchwald's protocol[43], the aryl−Pd bromide **4** was easily synthesized. To our delight, the subsequent anion exchange succeeded to afford the target Pd(II)−OCF₃ complex **5a**. Formation of **5a** was demonstrated by the ¹⁹F NMR, which showed a broad peak appeared at $\delta$ −30 ppm, although it completely decomposed in the NMR tube under room temperature overnight through the β-F elimination (Fig. 2A). During the preparation of our manuscript, Shen and co-workers reported the preparation of trifluoromethoxylated Pd(II) complexes, see ref. 44. The structure of **5a** was also confirmed by X-ray crystallography[44]. Furthermore, the thermal decomposition of **5a** favorably generated the aroyl fluoride product **2a**, suggesting that the Pd(II)−OCF₃ complex **5a** was the key intermediate in the reaction. The DFT calculation (see Supplementary Information for more details) also demonstrated that β-F elimination ($\Delta G^{\ddagger} = 8.4$ kcal/mol) was much more preferential than the direct reductive elimination ($\Delta G^{\ddagger} = 40.5$ kcal/mol) and explained the exclusive formation of aroyl fluoride.

The aryl−Pd trifluoromethesulfonate **6** was synthesized for the reaction course study of stoichiometric transformation by simultaneous ¹⁹F NMR characterization. To avoid the influence of heterogeneity, the soluble fluoride TASF[45] was used in place of inorganic fluorides. Pd(II)−OCF₃ complex **5b** was generated immediately after the injection of TFMS, and desired product **2y** was observed subsequently (Fig. 2B). A concomitant intermediate was also signified

in the spectrum at $\delta$ −47.06 (d, $J = 684.8$ Hz) and kept increasing with product **2y** over the reaction. Accordingly, the ³¹P NMR signal was a triplet and appeared at $\delta$ −20.08 (t, $J = 681.6$ Hz). The splitting pattern and slitting constant of around 680 Hz indicated a new P−F bond was formed.

Based on the structural analysis from ¹⁹F NMR, we believed the fluorine atom in the ligand was transferred from COF₂ gas[40], and the synthetic method of intermediate **7** was then proposed by treating Brettphos ligand with COF₂ which could be generated in situ from TFMS triggered by the catalytical amount of fluoride. As expected, the difluoro-Brettphos **7** was quantitatively obtained in ¹⁹F NMR and the precipitation generated after placing overnight was qualified for further spectroscopic characterization, including X-ray crystallography (Fig. 2C). Two identical fluorides were trans-positioned to each other at perpendicular axial and matched the NMR result that two F-atoms on the phosphorus center were equivalent. Meanwhile, COF₂ was reduced into CO. Encouraged by the stoichiometric experiments, the catalytic method for palladium/phosphine synergistic redox catalysis of fluorocarbonylation of potassium aryltrifluoroborate was investigated (see Supplementary Information for more details), and Xantphos was found to be the best ligand (Fig. 2D). The reason might be that once Xantphos was difluorinated, one vacant coordination site was generated and could be occupied by CO facilely, followed by the favorable migratory insertion to afford the aroyl−Pd complex, which was beneficial for the catalysis. Besides generating a new coordination site, Lewis acidic fluoro-phosphonium cation appended to a palladium

complex was also proved to promote CO insertion into a Pd–C bond[41,42] and lowered the activation barrier. To demonstrate the role of difluorophosphines in the reaction, the difluoro-Xantphos **L5** was synthesized and identified by X-ray crystallography; subsequently, it was used as a ligand and behaved similarly to Xantphos. An isotope tracing experiment was performed to verify the CO insertion instead of $COF_2$. When the complex **5a** was heated under a $^{13}CO$ atmosphere, the isotope-labeled aroyl fluoride was observed (Fig. 2E), suggesting the free CO molecule participated in the reaction. These results revealed the migratory insertion of CO was one of the critical steps in the catalytic cycle.

To disclose the full mechanism, especially for the synergistic redox of P(III/V)/ Pd(II/0) and the role of AgF, density functional theory (DFT) calculations were carried out by using Brettphos as the model ligand consistent with the control experiment in Fig. 3. DFT calculations were performed using Gaussian 09 at the SMD(1,4-Dioxane)-M06L/6-311+G(d,p)/Def2-TZVP(Pd,Ag) level of theory (see Supplementary Information for more details).

The dual catalysis enables the fluorocarbonylation reaction illustrated in the free energy profiles in Fig. 3. Our investigations indicate that the phosphine ligand indeed plays a bifunctional role, which can not only be a ligand to coordinate palladium but also can be a reactant with $COF_2$ to release CO. In Fig. 3A, the Brettphos ligand **L4** is facile to react with $COF_2$ via nucleophilic attack to form **INT1**, when the valence state of P would be oxidized from III to V valence. Then, fluoride ion migration of **INT1** occurs to afford five-coordinated **INT2** based on the P(V) center with the geometry of distorted trigonal bipyramidal configuration. The intermediate **INT2** then would be attacked by fluoride ions to form octahedral intermediate **INT3** or **INT3'**, which would release CO via **TS4** or **TS4'** to generate the *trans*-difluoro **INT4** or *cis*-difluoro **INT6**, respectively. And the *cis*-difluoro **INT6** could be isomerized to the stable *trans*-difluoro **INT4** supported by the isolated crystal structure **7**. Furthermore, F⁻ ligand dynamic coordination/dissociation during the transformation between **INT4** and **INT6** indicated the tuning effect of F⁻ anion for the spatial coordination of difluoro P(V) species. The turnover-limiting step of this phosphine cycle step has to overcome the 26.4 kcal/mol energy barrier that is accessible for a reaction that proceeds at 80 °C. Our calculated mechanism of the phosphine cycle rationalizes the generation of CO from $COF_2$, which is in agreement with our experimental observation in Fig. 2B.

For the Pd(II)/Pd(0) catalytic cycle, **L4** and **1a** were chosen as the starting materials in order to keep consistent with the zero point of P(III)-P(V) cycle shown in Fig. 3B. The Brettphos bound Pd(II) difluoro intermediate **INT8** was generated by favorable ligand exchange with Pd complex with exothermic −13.1 kcal/mol (1). And the additive AgF salt was introduced to activate **1a** through $S_EAr$ mechanism via **TS5** with a +3.8 kcal/mol low energy barrier to form the activated Ag-Ar intermediate **INT7** (2). The overall thermodynamic energy was exothermic −11.0 kcal/mol energy considering both ligand exchange (−13.1) and $S_EAr$ (+2.1). The formed Ag-Ar intermediate **INT7** could then react with Pd(II) species **INT8** through the key transmetallation, transferring the aryl group from Ag to Pd. This energy barrier of Ag assisted model via **TS6** was calculated to be favorable comparing other aryl group transfers in Supplementary Fig. S4, which was also supported by the study of aryl silver species[46–49] by Ritter et al. The following CO insertion step was initiated by Ar–Pd intermediate **INT10** and underwent three-membered ring transition state **TS8**, in which the Brettphos ligand displayed a slight ligand dissociation of $\eta^2$-aryl interaction to accept extra CO coordination. And this dynamic coordination of the biphenyl group in Brettphos ligand would re-bind to Pallidum in the next intermediate **INT13**. In the structure of **TS8**, the closest distance between Pd and $C_{Ar}$ on the ligand is 3.37 Å, while it turns into 2.53 Å and 2.63 Å in **INT11** and **INT13**, respectively. The following

reductive elimination was a low barrier step releasing activated Pd(0) species **INT13** and the final product **2a**. Then only *cis*-difluoro P(V) intermediate **INT6** was suggested to react Pd(0) via ligand exchange and oxidative addition regenerating catalyst species **INT8**. Note that the most stable *trans*-difluoro **INT4** cannot easily undergo oxidative addition via **TS11**, which is +4.6 kcal/mol relative energy higher than **TS10** with respect to the energy of zero point (**1a** +**L4**) (see Supplementary Information for details), probably due to the *trans*-effect contribution of F⁻ to hardly weaken C–F bond. So we speculate that *cis*-difluoro P(V) **INT6** is an active intermediate involved in the catalytic cycle. And the isomerization of *cis*- and *trans*- difluoro P(V) under F⁻ can control the reactivity to achieve this synergistic redox transformation. After the oxidative addition of the P–F bond to produce **INT15**, the second fluorine transfer tend to be facile through **TS12** with an even low energy barrier. Overall, the turnover-limiting step for Pd(II)/Pd(0) catalytic cycle would be the first oxidative addition via **TS10**, overcoming a reasonable 24.8 kcal/mol at 80 °C (Fig. 3B).

The substrate scope of the transformation was explored, and the generality of the reaction was covered in Fig. 4, demonstrating the efficient fluorocarbonylation of various (hetero)aryl/alkyl trifluoroborates, including some pharmaceutically relevant structures. The unstable acyl fluorines **2** were transformed into more valuable amides derivatives **3** immediately after the reaction following Manabe's protocol[26] due to the handy hydrolysis during chromatography purification. In general, for (hetero)aryl trifluoroborates, electronic and steric effects showed little influence on the reaction, while the electron-withdrawing groups and *meta*-substituents decreased the yields to some extent. Various substrates bearing electron-donating to electron-withdrawing substituents such as halides (**1a-1d, 1v**), alkyl and aryl groups (**1h, 1k, 1o, 1u**), alkoxyl (**1g, 1i, 1m**), ketone (**1n**), nitro (**1p**) and nitrile (**1q**) were successfully converted to the desired acyl fluorides with yields ranging from 50 to 99%. To our delight, most *ortho*-substituted substrates (**1e, 1j, 1l, 1o, 1t**) were well tolerated in this protocol and gave moderate yields (42 to 64%) with increasing steric hindrance. More importantly, the multi-substituted aryl trifluoroborates (**1d, 1j, 1o, 1v**) could be smoothly fluorocarbonylated. In addition, alkyl trifluoroborates (**1w-1ae**) were successfully converted to the corresponding desired alkyl fluorides with moderate yields under standard conditions. To prove the high compatibility of the reaction, the complicated pharmaceutical skeletons, Indometacin (**1af**), Fenofibrate (**1ag**), Gibberellic acid (**1ah**), Triclosan (**1ai**), Deoxycholic acid (**1aj**), and Estrone (**1ak**) derivatives were examined in the transformation and afforded the products (**2af, 2ag, 2ah, 2ai, 2aj, 2ak**) with moderate yields without touching the resting functional groups. Especially, **2aj** and **2ak** were obtained with isolated yields of 35 and 55%, respectively. These results showed the ability to allow the late-stage fluorocarbonylation of bioactive molecules and natural product derivatives.

## Discussion

In conclusion, we have developed the palladium/phosphine synergistic redox catalysis of fluorocarbonylation of potassium aryl/alkyl trifluoroborate with TFMS as a $COF_2$ source. The new method realized the direct conversion of potassium aryl/alkyl trifluoroborate to acyl fluorides, which allows the late-stage fluorocarbonylation of a variety of organic molecules and known drugs. Furthermore, the key intermediates of labile Pd(II)−$OCF_3$ complex **5a** and difluoro-Brettphos **7** were synthesized and spectroscopically characterized, including X-ray crystallography. In addition, the proposed reaction mechanism involving the interwoven catalytic cycles Pd(II)/(0) and P(III)/(V) was supported by experimental and computational studies and indicated that phosphine ligand plays a bifunctional role in the catalysis, which might stimulate the development of new reactions involving the synergistic

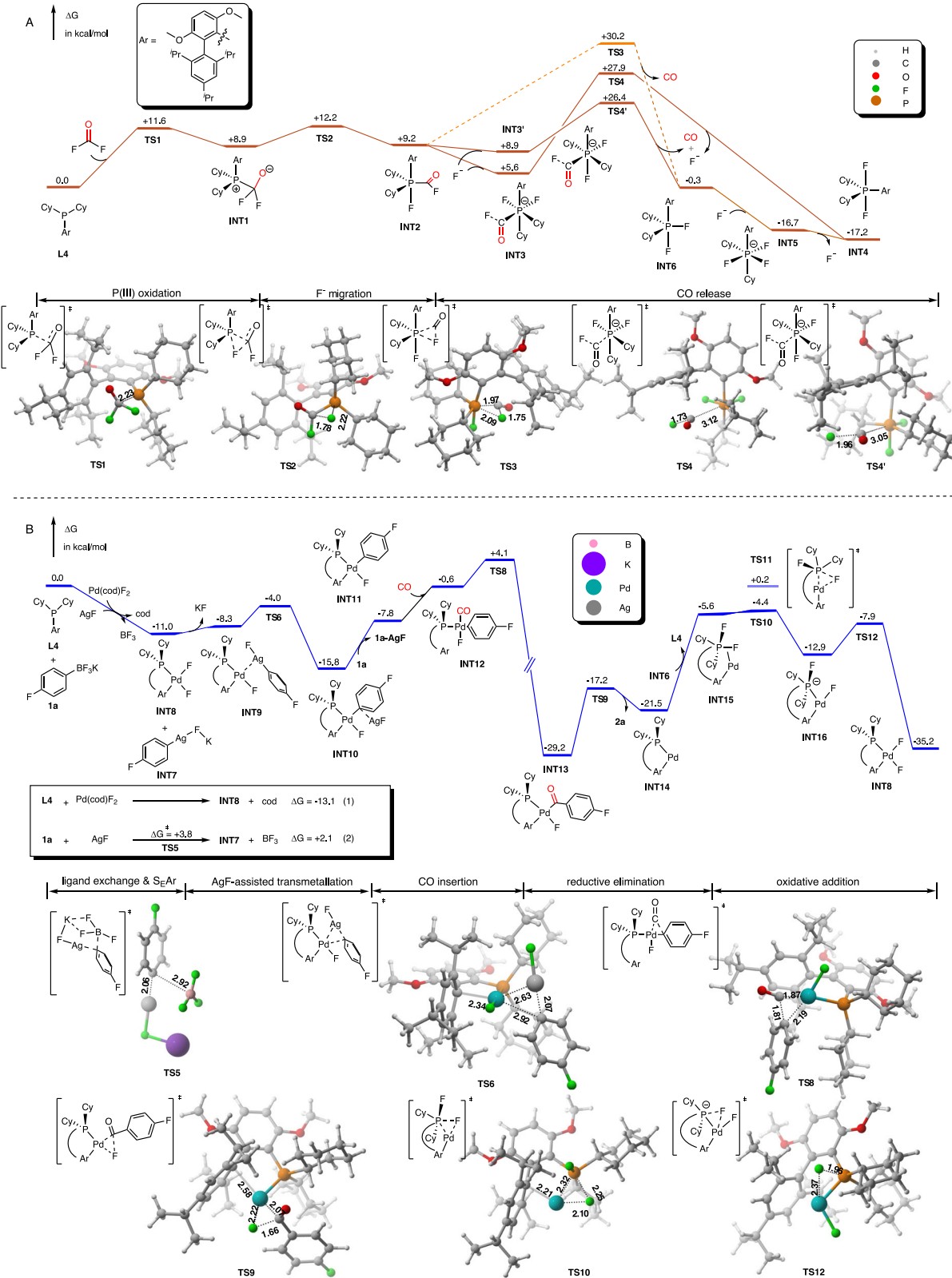

**Fig. 3 | Computational studies. A** Free energy profile for the catalytic cycle of the P(III)-P(V) mechanism releasing CO. Computed at the SMD(1,4-Dioxane)-M06L/6-311+G (d,p)//B3LYP-D3 /6–31G(d). **B** Free energy profile for the Pd(II)/Pd(0) redox of Pd(II)/Pd(0) and P(III)/P(V). Our studies provide a potential strategy to control the reducibility of P(V) species by the spatial coordination of fluoride ions, which might innovate more catalytic possibilities for widely used phosphine ligands.

catalytic cycle releasing product **2a**. Computed at the SMD(1,4-Dioxane)-M06L/6−311+G (d,p)/Def2-TZVP(Pd,Ag) //B3LYP-D3 /6–31G(d)/SDD(Pd,Ag).

## Methods

### General procedure for the synthesis of acyl fluorines

In an N₂ glovebox, to aryl fluoroboric acid potassium salt (**1**) (0.25 mmol, 1.00 equiv.), Pd(cod)Cl₂ (3.5 mg, 0.0125 mmol, 0.05

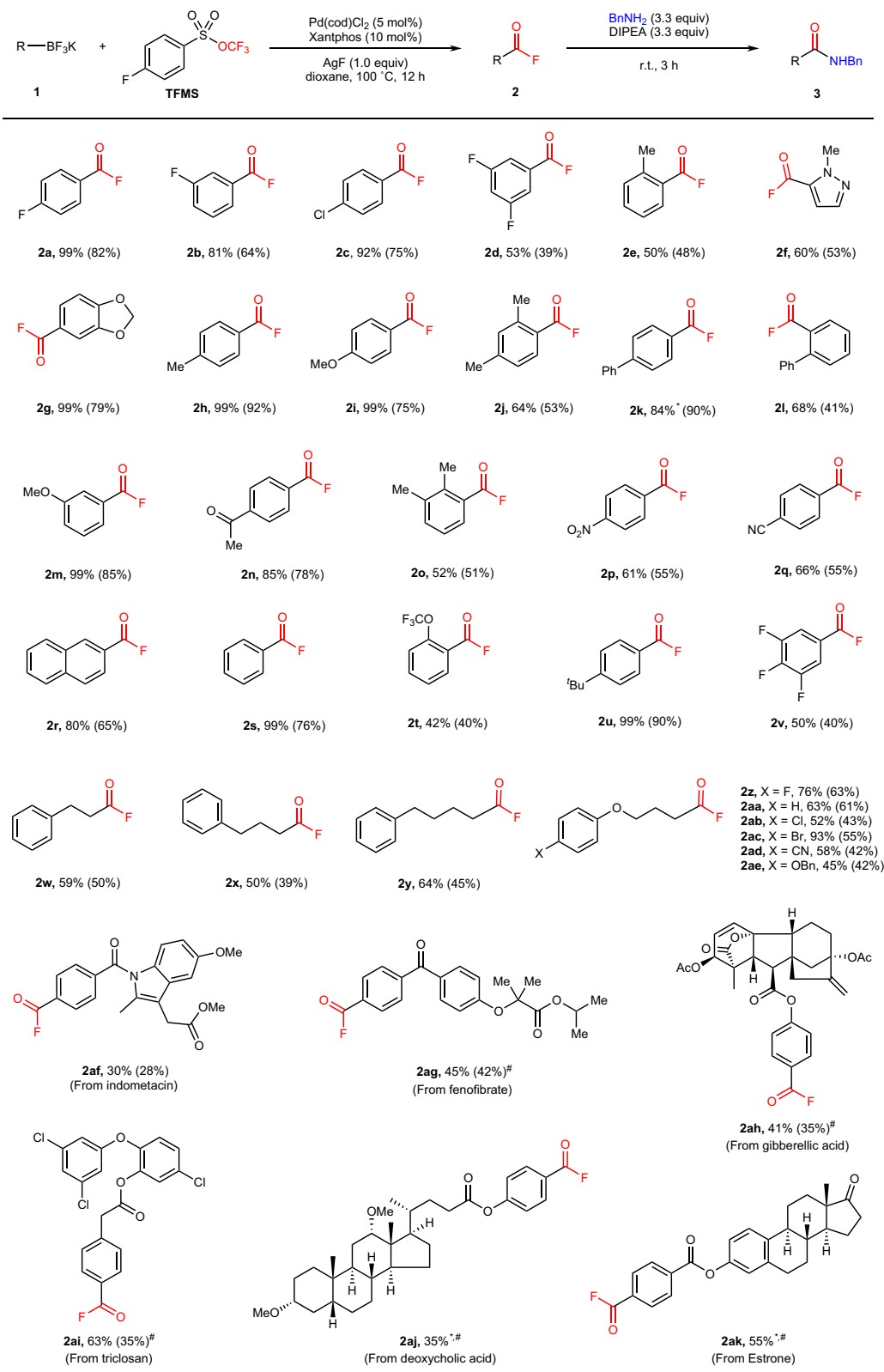

**Fig. 4 | Substrate scope for fluorocarbonylation of potassium aryl/alkyl tri-fluoroborate.** Reaction conditions: **1** (0.250 mmol, 1.0 equiv), TFMS (0.875 mmol, 3.5 equiv), Pd(cod)Cl$_2$ (0.0125 mmol, 5 mol%), Xantphos (0.025 mmol, 10 mol%), AgF (0.250 mmol, 1.0 equiv), dioxane (0.125 mol/L), 100 °C, 12 h, N$_2$ atmosphere. Yields of acyl fluorides **2** were determined by [19]F NMR with tribromofluoromethane or phenylsulfonyl fluoride as an internal standard unless otherwise noted. Yields of isolated amide derivatives **3** are given in parentheses. *Yields of isolated product **2**[#]. TFMS (4.0 equiv) was used.

equiv.), AgF (31.7 mg, 0.25 mmol, 1.00 equiv.) and Xantphos (14.5 mg, 0.025 mmol, 0.10 equiv.) in a 4.00 mL sealed vial tube were added 1,4-dioxane (2.00 mL) and TFMS (trifluoromethyl 4-fluor-obenzenesulfonate) (140.0 μL, 0.875 mmol, 3.50 equiv.). Then the sealed vial was taken outside the glovebox, and the reaction mixture was stirred for 12 h at 100 °C. The system was filtered and concentrated in vacuo. The residue was purified by preparative HPLC with acetoni-trile/water as an eluent.

### General procedure for the synthesis of amides derivatives

In an $N_2$ glovebox, to aryl/alkyl fluoroboric acid potassium salt (**1**) (0.25 mmol, 1.00 equiv.), $Pd(cod)Cl_2$ (3.5 mg, 0.0125 mmol, 0.05 equiv.), AgF (31.7 mg, 0.25 mmol, 1.00 equiv.) and Xantphos (14.5 mg, 0.025 mmol, 0.10 equiv.) in a 4.00 mL sealed vial tube were added 1,4-dioxane (2.00 mL) and TFMS (trifluoromethyl 4-fluor-obenzenesulfonate) (140.0 μL, 0.875 mmol, 3.50 equiv.). Then the sealed vial was taken outside the glovebox, and the reaction mixture was stirred for 12 h at 100 °C. The system was cooled to room tem-perature, bubbled with argon for 10 min, and added $BnNH_2$ (88.4 mg, 0.825 mmol, 3.3 equiv.) and diisopropylethylamine (DIPEA) (106.6 mg, 0.825 mmol, 3.3 equiv.), the reaction mixture was stirred for 3 h at room temperature. The system was filtered and concentrated in vacuo. The residue was purified by preparative TLC eluting with *n*-hexane/EA.

## Data availability

The authors declare that the data supporting this study are available within the article and supplementary information files (Supplementary Information, Supplementary Data 1). Crystallographic data for the structures reported in this article have been deposited at the Cam-bridge Crystallographic Data Centre under deposition numbers CCDC 2099433 (**5a**), 2099395 (**7**) and 2243163 (**L5**). Copies of the data can be obtained free of charge via https://www.ccdc.cam.ac.uk/structures/.

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

## Acknowledgements
This work was supported by the National Key Research and Development Program of China (2016YFA0602900, 2021YFA1500100), NFSC (21925105, 21890722, 92156017, 92156001), the Natural Science Foundation of Tianjin (Grant No. 18JCJQJC47000 and 19JCJQJC62300), NCC Fund (NCC2020FH07), "Frontiers Science Center for New Organic Matter," Nankai University (Grant Number 63181206), the Fundamental Research Funds for the Central Universities and the Haihe Laboratory of Sustainable Chemical Transformations.

## Author contributions
P.T. designed and directed the project. M.Z. and M.C. performed the experiments and analyzed the data. Q.P. directed the DFT calculations. T.W. and S.Y. conducted the DFT calculations. M.Z., M.C., T.W., Y.S., Q.P. and P.T. co-wrote the manuscript. All the authors discussed the results and commented on the manuscript.

## Competing interests
The authors declare no competing interests.
