## [Peer Review File · Nature Communications]

REVIEWER COMMENTS

Reviewer #1 (Remarks to the Author):

This manuscript from Tang and coworkers describes the development of palladium and phosphine synergistic redox catalysis of fluorocarbonylation of potassium aryl/alkyl trifluoroborate using TFMS reagent as an efficient source of COF₂.

This new reaction manifold allows the introduction of the COF onto aryl substrates in good to excellent yields. The methodology was also applied to alkyl derivatives. Finally, the authors clearly demonstrated the synthetic utility of the current method for the late-stage modification of drugs and natural products.

The mechanism of the reaction was also studied, and the key intermediates of labile Brettphos-PdII-OCF₃ complex and difluoro-Brettphos were synthesized and spectroscopically characterized including X-ray crystallography. A detailed reaction mechanism involving the synergistic redox catalytic cycles Pd(II)/(0) and P(III)/(V) was proposed and multifunction of phosphine ligand was identified based on ¹⁹F NMR, isotope tracing, synthetic, and computational studies.

To my opinion, this methodology is rather complementary to the existing ones and bring a clear added value to the current state of the art. Indeed, the synergistic redox catalytic cycles Pd(II)/(0) and P(III)/(V) is a clear added value. In addition, the TFMS, which was used as a trifluoromethoxylation reagent, has been employed as an efficient source of COF₂.

Hence, I would recommend this manuscript for publication in Nature Communications.

Questions:

- 1) what's the byproduct of this transformation?
- 2) Did the author observe the fluorine peaks of cis-difluoro INT6 by the ¹⁹F NMR?

Reviewer #2 (Remarks to the Author):

In this manuscript, the authors reported fluorocarbonylation of potassium aryl/alkyl trifluoroborate with the use of trifluoromethyl arylsulfonate (TFMS) as COF₂ source through the palladium/phosphine synergistic redox catalysis. The substrate scope of the reaction was wide, and

the yields were acceptable. The control experiments and DFT calculations led to a plausible mechanism proposed, which involved the synergistic redox of Pd(II)/Pd(0) and P(III)/P(V). The current reaction could be a complementary method for fluorocarbonylation, and also provided an inspiration for palladium catalysis and the use of phosphines. The reviewer recommends accepting the manuscript for publication in Nature Communications after the revision of the following points.

1. It is better to give the structure of difluoro-Xantphos in the manuscript or SI.
2. The structure of I-1 was wrongly drawn in Figure 1.
3. How about reactivity of other heteroaryl trifluoroborate besides one example in Figure 4?
4. There are lots of format errors in the manuscript.

Reviewer #3 (Remarks to the Author):

As acyl fluorides are the very important synthons, the development of methods for practical preparation of acyl fluorides has received a great attention in organic chemistry community and big progress has been made recently. In this manuscript, Tang and co-workers described a new method for the conversion of potassium aryl/alkyl trifluoroborate to acyl fluorides. This reaction proceeded through the palladium/phosphine synergistic redox catalysis of fluorocarbonylation of potassium aryl/alkyl trifluoroborate using trifluoromethyl arylsulfonate (TFMS) as a COF₂ source. This synthetic method was applied for the late stage fluorocarbonylation of a variety of organic molecules and

known drugs. The key intermediates of labile PdII-OCF₃ complex and difluoroBrettphos were synthesized and characterized. Furthermore, the proposed reaction mechanism was investigated in detail by experimental and computational studies. General speaking, the developed method for synthesis of acyl fluorides is not direct and practical. But the reaction involving the synergistic redox of Pd(II)/Pd(0) and P(III)/P(V) is breakthrough in organometallic chemistry and fluorine chemistry. In view of the novelty of the reaction design and proposed reaction mechanism, this manuscript was recommended for publication in Nature Communication when the following comments were addressed.

- 1) The best ligand for this transformation is Xantphos. Why was Brettphos used as ligand instead of Xantphos for synthesis of PdII-OCF₃ complex (5a) and aryl-Pd trifluoromethanesulfonate (6)?
- 2) Regarding to Figure 2C, was CO detected?
- 3) Regarding to reaction design (Figure 1), Could Intermediates I-2 and I-3 be isolated and characterized?

4) Regarding to Figure 2A, the authors stated in the text that 5a was only detected by ^{19}F NMR. But the X-ray structure of 5a was shown in Figure 2A. Please confirm it.

Reviewer #4 (Remarks to the Author):

The authors present an interesting, new catalytic procedure to obtain acyl fluorides, which are of potential relevance as synthons for subsequent synthesis of novel fluorinated drugs. The study demonstrates that the catalytic process is compatible with a large substrate scope. One of the main strengths of the work is the detailed mechanistic study of the process that comprises stoichiometric transformations of key intermediates, catalytic experiments, labelling experiments, and computational (DFT) studies. In my opinion the main trends of the mechanism are captured by these set of tools. However, regarding the atomistic description obtained by DFT calculations, there are some inconsistencies that, in my opinion, need to be solved before the manuscript is accepted for publication:

1. In page 8, at the beginning of the discussion of "Pd" catalytic cycle, the authors assume that "INT8 species", which consists of the coordinated ligand, is the "reactive species". In my opinion, this is quite reasonable. However, when discussing the energy profiles, it seems that INT8 is set to the zero-energy reference, while in "phosphine" cycle the free ligand is set to zero-energy. This would not be conceptually correct because the two catalytic cycles operate simultaneously in one pot reaction. The two catalytic cycle should be discussed using the same species as the zero of energy.

2. The discussion of the last part of the mechanism, the regeneration of the active Pd(II) species (page 9), is not fully convincing, and I would qualify it as somewhat superfluous. Actually, this is key part of the mechanism because it corresponds to the rate-determining process of the catalytic cycle. Firstly, the transfer of the second fluoride to the coordination sphere of Pd is not described. Secondly, the energy reference for the difluoro P(V) intermediate is not clear. It seems from Figure 3 B that the cis-difluoro P(V) (INT6) is responsible for the Pd(0) oxidative addition to Pd(II). In the free-energy barrier calculation, do the authors count the energy cost of the isomerization from the most stable trans-difluoro P(V) (INT4) isomer? As argued in previous point, it is not clear whether the two cycles use the same zero-energy point as reference. Surprisingly, the energy difference between the oxidative addition transition states for cis and trans paths is only 4.6 kcal/mol, while the energy difference between cis and trans reactants is 16.8 kcal/mol.

Besides these comments, the authors may consider some additional minor revisions during preparation of the next version:

- In Figure 1, the intermediate I-1 has an additional H atom drawn attached to the P atom.
- In page 3, while referring to the mechanism, the authors state that “we hoped that phosphine ligand L4 reacts...”. The term “hoped” is not appropriate for a scientific document.
- In Figure 3 B, the release of BF₃ at the early steps of the reaction should be drawn in the energy profile.

Revised Manuscript Submission NCOMMS-22-47540-T

March 14, 2023

Dear referees,

Thank you very much for your professional comments and advice. After learning from those comments, we have done some experiments to address your concerns.

In the revised manuscript, we have added additional experiments including the synthesized difluoro-Xantphos (L5) and identified its structure by X-ray crystallography. We have commented in detail on all comments by the reviewers (reviewer comments unchanged below). We agree with most comments and are pleased to provide additional results, which we believe address your concerns. All the changes in the revised manuscript and SI are highlighted in yellow color.

Our answers are listed as follows:

Reviewer Comments:

Reviewer #1 (Remarks to the Author):

This manuscript from Tang and coworkers describes the development of palladium and phosphine synergistic redox catalysis of fluorocarbonylation of potassium aryl/alkyl trifluoroborate using TFMS reagent as an efficient source of COF₂.

This new reaction manifold allows the introduction of the COF onto aryl substrates in good to excellent yields. The methodology was also applied to alkyl derivatives. Finally, the authors clearly demonstrated the synthetic utility of the current method for the late-stage modification of drugs and natural products.

The mechanism of the reaction was also studied, and the key intermediates of labile Brettphos-PdII-OCF₃ complex and difluoro-Brettphos were synthesized and spectroscopically characterized including X-ray crystallography. A detailed reaction mechanism involving the synergistic redox catalytic cycles Pd(II)/(0) and P(III)/(V) was proposed and multifunction of phosphine ligand was identified based on ¹⁹F NMR, isotope tracing, synthetic, and computational studies.

To my opinion, this methodology is rather complementary to the existing ones and bring a clear added value to the current state of the art. Indeed, the synergistic redox catalytic cycles Pd(II)/(0) and P(III)/(V) is a clear added value. In addition, the TFMS, which was used as a trifluoromethoxylation reagent, has been employed as an efficient source of COF₂.

Hence, I would recommend this manuscript for publication in Nature Communications.

Questions:

1) what's the byproduct of this transformation?

We appreciate the reviewer's comment. The main by-products are protonation products and dimers.

2) Did the author observe the fluorine peaks of cis-difluoro INT6 by the ^{19}F NMR?

We appreciate the reviewer's comment. We had only observed the fluorine peaks of trans-difluoro INT5 by the ^{19}F NMR in this reaction system, and we had not observed the fluorine peaks of cis-difluoro INT6.

Thank you very much for your professional comments!

Reviewer #2 (Remarks to the Author):

In this manuscript, the authors reported fluorocarbonylation of potassium aryl/alkyl trifluoroborate with the use of trifluoromethyl arylsulfonate (TFMS) as COF₂ source through the palladium/phosphine synergistic redox catalysis. The substrate scope of the reaction was wide, and the yields were acceptable. The control experiments and DFT calculations led to a plausible mechanism proposed, which involved the synergistic redox of Pd(II)/Pd(0) and P(III)/P(V). The current reaction could be a complementary method for fluorocarbonylation, and also provided an inspiration for palladium catalysis and the use of phosphines. The reviewer recommends accepting the manuscript for publication in Nature Communications after the revision of the following points.

1) It is better to give the structure of difluoro-Xantphos in the manuscript or SI.

We appreciate the reviewer's comment. We have synthesized difluoro-Xantphos (L5) and identified its structure by X-ray crystallography. we have added this information in the revised manuscript and SI.

2) The structure of I-1 was wrongly drawn in Figure 1.

We appreciate the reviewer's comment. We have modified the structure of I-1 in Figure 1 to remove excess "H" in the revised manuscript.

3) How about reactivity of other heteroaryl trifluoroborate besides one example in Figure 4?

We appreciate the reviewer's comment. We tried other heteroaryl borates such as furans, thiophenes, pyridines, benzofurans and benzothiophenes, but unfortunately, we did not obtain the corresponding acyl fluoride products, the main by-products were dimers and protonation products.

4) There are lots of format errors in the manuscript.

We appreciate the reviewer's comment. we had rechecked and revised the format errors that appeared in the article, and the modified parts have been highlighted in yellow. For example, "iPr" has been revised as "iPr", "Pd(II)" and "PdII" have been uniformly revised as "Pd(II)".

Thank you very much for your professional comments!

Reviewer #3 (Remarks to the Author):

As acyl fluorides are the very important synthons, the development of methods for practical preparation of acyl fluorides has received a great attention in organic chemistry community and big progress has been made recently. In this manuscript, Tang and co-workers described a new method for the conversion of potassium aryl/alkyl trifluoroborate to acyl fluorides. This reaction proceeded through the palladium/phosphine synergistic redox catalysis of fluorocarbonylation of potassium aryl/alkyl trifluoroborate using trifluoromethyl arylsulfonate (TFMS)

as a COF₂ source. This synthetic method was applied for the late stage fluorocarbonylation of a variety of organic molecules and known drugs. The key intermediates of labile PdII-OCF₃ complex and difluoroBrettphos were synthesized and characterized. Furthermore, the proposed reaction mechanism was investigated in detail by experimental and computational studies. General speaking, the developed method for synthesis of acyl fluorides is not direct and practical. But the reaction involving the synergistic redox of Pd(II)/Pd(0) and P(III)/P(V) is breakthrough in organometallic chemistry and fluorine chemistry. In view of the novelty of the reaction design and proposed reaction mechanism, this manuscript was recommended for publication in Nature Communication when the following comments were addressed.

1) The best ligand for this transformation is Xantphos. Why was Brettphos used as ligand instead of Xantphos for synthesis of PdII-OCF₃ complex (5a) and aryl-Pd trifluoromethanesulfonate (6)?

We appreciate the reviewer's comment. According to the literature of Buchwald and co-workers, they had successfully used Brettphos to synthesize Pd-F complex, because OCF₃ group has many similarities with F fluoride ions, we first chose Brettphos as a raw material for the synthesis of Pd(II)-OCF₃ complex. Later, we also tried to synthesize Pd(II)-OCF₃ complex using Xantphos as a raw material, but unfortunately did not succeed in obtaining the ideal product.

2) Regarding to Figure 2C, was CO detected?

We appreciate the reviewer's comment. The formation of carbon monoxide was detected by a carbon monoxide detector tube, and the specific operation was attached to the Supporting Information scheme S1.

3) Regarding to reaction design (Figure 1), Could Intermediates I-2 and I-3 be isolated and characterized?

We appreciate the reviewer's comment. We tried to use equivalent palladium to participate in the reaction and quench the reaction after 2 hours. We observed the formation of a small amount of acyl fluoride product, unfortunately, no signal of Intermediates I-2 or I-3 was observed in ¹⁹F NMR. However, referring to Buchwald's method, we obtained Pd(II)-F complex INT10 indicated by the ¹⁹F NMR at δ -119.81 (d, J = 117.4 Hz). And then the Pd(II)-F complex INT10 was added to the reaction system, which potassium trifluoro(phenyl)borate as the reaction substrate. Two different acyl fluoride products 2a and 2s was determined by the ¹⁹F NMR and GC-MS, which demonstrate that Pd(II)-F complex INT10 is a reactive species.

4) Regarding to Figure 2A, the authors stated in the text that 5a was only detected by ¹⁹F NMR. But the X-ray structure of 5a was shown in Figure 2A. Please confirm it.

We appreciate the reviewer's comment. The structure of 5a was confirmed by X-ray crystallography in low temperatures, although it easily decomposes at room temperature. And we have revised the description of 5a in the main text and highlighted it in yellow.

Thank you very much for your professional comments!

Reviewer #4 (Remarks to the Author):

The authors present an interesting, new catalytic procedure to obtain acyl fluorides, which are of potential relevance as synthons for subsequent synthesis of novel fluorinated drugs. The study demonstrate that the catalytic process is compatible with a large substrate scope. One of the main strengths of the work is the detailed mechanistic study of the process that comprises stoichiometric transformations of key intermediates, catalytic experiments, labelling experiments, and computational (DFT) studies. In my opinion the main trends of the mechanism are captured by these set of tools. However, regarding the atomistic description obtained by DFT calculations, there are some inconsistencies that, in my opinion, need to be solved before the manuscript is accepted for publication:

1) In page 8, at the beginning of the discussion of "Pd" catalytic cycle, the authors assume that "INT8 species", which consists of the coordinated ligand, is the "reactive species". In my opinion, this is quite reasonable. However, when discussing the energy profiles, it seems that INT8 is set to the zero-energy reference, while in "phosphine" cycle the free ligand is set to zero-energy. This would not be conceptually correct because the two catalytic cycles operate simultaneously in one pot reaction. The two catalytic cycle should be discussed using the same species as the zero of energy.

We appreciate the reviewer's comment. As shown in the revised manuscript, we unified the zero point of the two cycles. In the Pd(II)-Pd(0) cycle, since **1a** is one of the reactants, in order to enable **1a** can be shown in the figure and to facilitate the description of the subsequent reactions, we used **1a** and **L4** as the zero point in the Pd(II)-Pd(0) cycle.

2) The discussion of the last part of the mechanism, the regeneration of the active Pd(II) species (page 9), is not fully convincing, and I would qualify it as somewhat superfluous. Actually, this is key part of the mechanism because it corresponds to the rate-determining process of the catalytic cycle. Firstly, the transfer of the second fluoride to the coordination sphere of Pd is not described. Secondly, the energy reference for the difluoro P(V) intermediate is not clear. It seems from Figure 3 B that the cis-difluoro P(V) (INT6) is responsible for the Pd(0) oxidative addition to Pd(II). In the free-energy barrier calculation, do the authors count the energy cost of the isomerization from the most stable trans-difluoro P(V) (INT4) isomer? As argued in previous point, it is not clear whether the two cycles use the same zero-energy point as reference. Surprisingly, the energy difference between the oxidative addition transition states for cis and trans paths is only 4.6 kcal/mol, while the energy difference between cis and trans reactants is 16.8 kcal/mol.

We appreciate the reviewer's comment. Firstly, we found the transition state of the second fluorine transfer and showed it in Figure 3B. Its structure is shown in the Figure in the revised manuscript. Secondly, we calculated that the absolute energy of TS10 is 21.4 kcal/mol lower than that of TS11. The energy difference between the two in Figure 3B is 4.6 kcal/mol, which is obtained after taking into account the energy converted between INT4 and INT6.

Besides these comments, the authors may consider some additional minor revisions during preparation of the next version:

- In Figure 1, the intermediate I-1 has an additional H atom drawn attached to the P atom.
- In page 3, while referring to the mechanism, the authors state that "we hoped that phosphine ligand L4 reacts...". The term "hoped" is not appropriate for a scientific document.

- In Figure 3 B, the release of BF₃ at the early steps of the reaction should be drawn in the energy profile.

We appreciate the reviewer's comment. we have modified the structure of I-1 in Figure 1 to remove excess "H", and replaced "hoped" with "envisioned ". We have modified Figure 3B to add the release of BF₃ in addition to the data changes. And we have rechecked and revised the other format errors that appeared in the article, and the modified parts have been highlighted in yellow. For example, "iPr" has been revised as "iPr", "Pd(II)" and "PdII" have been uniformly revised as "Pd(II)".

Thank you very much for your professional comments!

Thank you for your consideration of this manuscript.

Sincerely,

Pingping Tang

REVIEWER COMMENTS

Reviewer #1 (Remarks to the Author):

Most of the comments from reviewers were addressed. Therefore, I recommend its publication in Nature Communications.

Reviewer #2 (Remarks to the Author):

In the revised manuscript, the issues raised by the reviewers have been addressed. The current manuscript is suitable for publication in Nature Communications.

Reviewer #3 (Remarks to the Author):

My comments were fully addressed. This revised manuscript was recommended for publication.

Reviewer #4 (Remarks to the Author):

The authors have followed my suggestion to use the same zero-energy for the two cycles in order to allow a more straightforward energy comparison. Nevertheless, the new discussion (lines 146 to 154) has become very confusing because it does not seem consistent with the E profile in Figure 3D: (1) "INT8 was generated by favourable ligand exchange", but the energy difference is +6.1 in Figure 3B, (2) the energy of INT7 is not provided in Figure 3, and it is not integrated in the overall E profile, (3) "The overall energy was favourable with 6.1 kcal/mol energy considering both ligand exchange (-14.1) and SEAr mechanism (20.1)", but this is not illustrated in Figure 3B, (4) in lines 160-161, the formation of Pd(0) intermediate INT3 is qualified as "facile step", while the process is endergonic with a low reverse energy barrier, (5) in the rate determining step (TS10 and TS11), it is not clearly explained how the energies of P(V) species (trans and cis) are incorporated in the catalytic cycle (set to zero-energy or consider their relative energy respect to 1a?), what is a key point in the mechanistic description. In my opinion, the new description of the DFT-derived mechanism is less

clear than in previous version, and therefore, I can not support publication of the manuscript in its current form.

Revised Manuscript Submission NCOMMS-22-47540A

May 30, 2023

Dear referees,

Thank you very much for your professional comments and advice. After learning from those comments, we have revised our manuscript to address your concerns.

In the revised manuscript, we have revised the computational parts and replied the concerns by reviewer 4. All the changes in the revised manuscript and SI are highlighted in yellow color.

Our answers are listed as follows:

Reviewer #4 (Remarks to the Author):

The authors have followed my suggestion to use the same zero-energy for the two cycles in order to allow a more straightforward energy comparison. Nevertheless, the new discussion (lines 146 to 154) has become very confusing because it does not seem consistent with the E profile in Figure 3D: (1) “INT8 was generated by favourable ligand exchange”, but the energy difference is +6.1 in Figure 3B, (2) the energy of INT7 is not provided in Figure 3, and it is not integrated in the overall E profile, (3) “The overall energy was favourable with 6.1 kcal/mol energy considering both ligand exchange (-14.1) and SEAr mechanism (20.1)”, but this is not illustrated in Figure 3B, (4) in lines 160-161, the formation of Pd(0) intermediate INT3 is qualified as “facile step”, while the process is endergonic with a low reverse energy barrier, (5) in the rate determining step (TS10 and TS11), it is not clearly explained how the energies of P(V) species (trans and cis) are incorporated in the catalytic cycle (set to zero-energy or consider their relative energy respect to 1a?), what is a key point in the mechanistic description. In my opinion, the new description of the DFT-derived mechanism is less clear than in previous version, and therefore, I can not support publication of the manuscript in its current form.

We appreciate the reviewer’s professional comments that help us to fix computational errors. And we double-checked all calculated results and revised the manuscript accordingly to make this part clear.

Nevertheless, the new discussion (lines 146 to 154) has become very confusing because it does not seem consistent with the E profile in Figure 3D: (1) “INT8 was generated by favourable ligand exchange”, but the energy difference is +6.1 in Figure 3B, (2) the energy of INT7 is not provided in Figure 3, and it is not integrated in the overall E profile, (3) “The overall energy was favourable with 6.1 kcal/mol energy considering both ligand exchange (-14.1) and SEAr mechanism (20.1)”, but this is not illustrated in Figure 3B

Reply: In order to adjust the same zero-energy between Pd(II)/Pd(0) and P(III)/P(V), we have to combine the independent two processes of the ligand exchange (eq. 1) and SEAr mechanism (eq. 2)

together. The total thermodynamic energy is exothermic -11.0 kcal/mol. Actually, the energy of corresponding intermediates had been included in the original Figure 3B. (eg. we add INT8 and INT7 together for -11.0 kcal/mol relative energy). Because of this energy zero point, all the intermediates and transition states would be corrected comparing with previous submissions. Computational errors had been fixed or recalculated. We have added the eq 1 and 2 into Figure 3B to make this issue clear.

(4) in lines 160-161, the formation of Pd(0) intermediate INT3 is qualified as “facile step”, while the process is endergonic with a low reverse energy barrier,

Reply: We have revised this part to “low energy barrier step”.

(5) in the rate determining step (TS10 and TS11), it is not clearly explained how the energies of P(V) species (trans and cis) are incorporated in the catalytic cycle (set to zero-energy or consider their relative energy respect to 1a?), what is a key point in the mechanistic description.

Reply: Thanks for the comment. The relative energy of **TS10** and **TS11** had been calculated based the new zero point (**1a** + **L4**) suggested by this reviewer. Their relative energy would be 4.6 kcal/mol. And if we set to zero-energy based on **INT14**, the energy profile would be as shown below, suggesting the *trans*-difluoro P(V) (**INT4**) is inert for the next P-F oxidative addition. To ensure the rationality of the calculation, we also validated the reaction process using the simplified model and obtained the same conclusion that *cis*-difluoro P(V)INT6 (or **INT6-S**) is the plausible active intermediate in the P-F oxidative addition.

Figure R1. P-F oxidative addition with **INT14** as zero point for full model and simplified model.

Table R1. Relative energy differences of intermediate and transition state for full and simplified model.

		Full model	Simplify Model	
Intermediate relative energy difference				
$\Delta\Delta G$	INT4	-16.9	INT4-S	-17.3
(kcal/mol)	INT6	0.0	INT6-S	0.0
Transition state relative energy difference				
$\Delta\Delta G^\ddagger$	TS10	0.0	TS10-S	0.0
(kcal/mol)	TS11	+4.6	TS11-S	+12.5

We appreciate the reviewer's professional comments to improve our paper.

Thank you for the editor's consideration of this manuscript.

Sincerely,

Pingping Tang

REVIEWERS' COMMENTS

Reviewer #4 (Remarks to the Author):

The new revision of the authors have accounted my previous concerns, and therefore, I can support publication of the manuscript.